# The Time-Dependent Effect in Ultra High-Performance Concrete According to the Curing Methods

**DOI:** 10.3390/ma15145066

**Published:** 2022-07-21

**Authors:** Kwangmo Lim, Kyongchul Kim, Kyungtaek Koh, Gumsung Ryu

**Affiliations:** 1Korean Peninsula Infrastructure Special Committee, Korea Institute of Civil Engineering and Building Technology, Goyang-si 10223, Korea; limkm@kict.re.kr (K.L.); kim6069@kict.re.kr (K.K.); ktgo@kict.re.kr (K.K.); 2Department of Structural Engineering Research, Korea Institute of Civil Engineering and Building Technology, Goyang-si 10223, Korea

**Keywords:** ultra-high-performance concrete (UHPC), precast production, bridge girders, concrete curing, long-term performance

## Abstract

Ultra-high-performance concrete (UHPC) is required to develop multifunctional concrete structures such as long-span bridges. During the construction of long-span bridges, girders exhibit significant differences in age because they use different curing days in the precast process. In this study, the performances of UHPC were compared when subjected to long-term storage under various conditions after 3-day steam curing. At 365 days, the compressive strength of steam curing is 197 MPa, moist is 191 MPa, and the air is 169 MPa. Based on these differences, prediction models were proposed for long-term performances. Furthermore, the development characteristics of compressive strength, modulus of elasticity (MOE), and flexural strength until 365 days of age were analyzed under air, moist, and steam conditions. Steam curing exhibited the highest level of strength development while air curing showed the lowest. Flexural strength showed no significant difference depending on age because steel fibers were mixed with UHPC; they significantly contributed to flexural performance. The results would contribute to recognizing differences in strength between members at sites where UHPC is applied and to managing high-quality structures constructed using precast members. These research results are expected to contribute to efficient member production and process management during the construction of large structures such as super-long-span bridges.

## 1. Introduction

In the construction market, high-performance concrete materials such as ultra-high-performance concrete (UHPC) have been developed to realize multifunctional concrete structures. Concrete performance evaluation needs to be sufficiently performed to increase UHPC commercialization. The time-dependent performance analysis of concrete should be evaluated in parallel because the lifespan of a structure can be increased using UHPC.

UHPC is produced using a precast process to maintain high quality. The precast production method exhibits high efficiency when considering the continuous construction of bridge girders as a representative example. In this precast process, high strength is secured by performing steam curing during girder production. However, the girders produced by this process did not move to construction sites upon the completion of steam curing. Those members would be stored for several days under different conditions depending on the site situation. Some of these girders are stored in a steam curing room or under moist conditions to maintain quality, whereas others are stored under air-dry conditions. During the construction of long-span bridges, girders can exhibit significant differences in age because they have different curing days. That is, each member may exhibit different strength characteristics at the time of being applied to an actual site despite having the same mix properties and strengths. Thus, there is a need to determine the long-term behavior of UHPC under each curing condition.

Research has been continuously conducted to predict the strength of UHPC. Marani et al. (2020) investigated a novel framework for predicting the compressive strength of UHPC using a tabular generative adversarial net model with 810 experimental data points from the literature [1]. Zhang et al. (2017), Abuodeh et al. (2020), and Nguten et al. (2022) suggested an artificial neural network (ANN) model to predict the compressive strength of the UHPC [2,3,4]. In these studies, deep machine learning techniques were used to determine the compressive strength in a complex cementitious material matrix.

Considering the importance of the strength development characteristics of UHPC according to the curing method, some researchers conducted experimental studies of various aspects. Mohammed et al. (2021) conducted an experimental study using the mixing contents of steel fibers as a variable in steam and water curing conditions [5]. Zhu et al. (2020) evaluated the performance of UHPC according to the constraint under air and steam curing conditions [6].

However, since concrete has used various materials and fibers and shows inhomogeneous characteristics, it is difficult to identify the material clearly. In particular, it is difficult to accumulate sufficient data for long-term behavior evaluation because long-term experiments are required.

Therefore, this study analyzed the long-term performance of UHPC according to the curing method. The performance was evaluated under air curing conditions to simulate the precast member being stored in the open area after being manufactured. The performance of each curing method was comparatively investigated, including steam curing, which is typically used in the manufacture of precast members, and moist curing, which is generally applied to the curing of concrete. For a 150 MPa UHPC, the strength, modulus of elasticity (MOE), and flexural strength until 365 days of age were analyzed experimentally. In addition, the long-term structural performance of the UHPC was predicted and used as primary data to identify long-term behavior. The results are expected to contribute to recognizing differences in strength between members at sites when the UHPC is applied and managing high-quality precast members.

## 2. Literature Review

### 2.1. Prediction of Compressive Strength

The American Concrete Institute (ACI) proposed a compressive strength development prediction model according to the time history based on an empirical formula given by Equation (1) [7,8].
(1)$${\left(f_{c}^{\prime }\right)}_{t}=\frac{t}{\alpha +\beta t}{\left(f_{c}^{\prime }\right)}_{28}$$

According to the ASTM C511, Equation (1) requires a moist curing temperature of 23 ± 2 °C with a relative humidity of 50%, and that requires a curing temperature of 100 °C for steam curing. In this Equation (1), t means days of the concrete age. The ${\left(f_{c}^{\prime }\right)}_{t}$ indicates the compressive strength at the day of t, and ${\left(f_{c}^{\prime }\right)}_{28}$ is that of 28 days. This equation applies to normal strength concrete (NSC) with compressive strength from 27.6 to 31.0 Mpa when Type 1 cement is used. For Equation (1), α = 4.0 and β = 0.85 are used under general curing conditions and α = 1.0 and β = 0.95 are steam curing.

The British Standards Institution (BS) presents the compressive strength of concrete according to age as Equation (2) [9].
(2)$$\begin{array}{c} f_{cm}\left(t\right)=\beta_{cc}\left(t\right)f_{cm}\\ \beta_{cc}\left(t\right)=\exp[s[1-{\left(\frac{28}{t}\right)}^{\left(1/2\right)}]] \end{array}$$

Equation (2) is based on concrete prepared in accordance with EN12390 under the curing condition of 20 °C, and it comprises the age and cement type. In the equation, $f_{cm}\left(t\right)$ represents the concrete strength at age t, and $f_{cm}$ represents the strength at 28 days. The cement types mentioned in the equation are divided into CEM 32.5N and CEM 52.5R according to the mix strength. In addition, 0.2 is used as the coefficient *s* for concrete with a compressive strength of 60 Mpa or higher. For this model, BS limits the application range to NSC and HSC and provides a range from 12 to 90 Mpa.

The Comité Européen du Béton-International Federation for Structural Concrete (CEB-FIB) code allows Equation (2) to be used for predicting the compressive strength of concrete based on age [10]. The specimen for this prediction model should be prepared in accordance with ISO 1920-3 under the curing condition of 20 °C. Although the same equation is used, the CEB code specifies the range of up to 120 Mpa for concrete, unlike the BS code.

The BS code predicts the strength of concrete under heat curing conditions using Equation (3) [9].
(3)$$f_{cm}\left(t\right)=f_{cmp}+\frac{f_{cm}-f_{cmp}}{\log\left(28-t_{p}+1\right)}\log\left(t-t_{p}+1\right)$$

Equation (3) predicts the strength of a precast member at age *t* using its strength at the end of steam curing. In this equation, $t_{p}$ represents the age at the end of steam curing, and $f_{cmp}$ represents the strength at that time.

The creep, drying shrinkage, and autogenous shrinkage should significantly influence the long-term behavior of concrete. The long-term behavior of UHPC has been actively researched, considering various variables, and several of these studies have investigated the creep and shrinkage effects [11,12,13]. In addition, studies that investigate compressive strength development characteristics based on the age of concrete can be found in some previous literature [14,15,16]. However, there are few studies on the long-term monitoring of the compressive strength of UHPC. Although concrete tends to increase in strength with increasing age, it is a theme that must be studied to understand the behavior of concrete clearly. Therefore, in this study, the enhancement of the prediction model has been suggested in the ACI, CEB, and BS criteria by comparing it with the experimental results of UHPC.

### 2.2. Prediction of the Modulus of Elasticity

The ACI standard predicts the modulus of elasticity (MOE) using Equation (4).
(4)$$E_{ct}=g_{ct}{\left[w^{3}{\left(f_{c}^{\prime }\right)}_{t}\right]}^{1/2}$$

For MOE, g*_ct_* is used for 0.043, and *w* represents the unit weight of concrete. This MOE is affected by the change in concrete strength based on the age in Equation (1) [7].

The BS code provides Equation (5) that uses MOE at 28 days of age. The coefficient of the equation is obtained from the strength of concrete at 28 days of age and the concrete strength at a certain age. *F_cm_*(*t*) can be obtained from Equation (2).
(5)$$E_{cm}\left(t\right)={\left(\frac{f_{cm}\left(t\right)}{f_{cm}}\right)}^{0.3}E_{cm}$$

Equation (6) is the MOE prediction formula suggested by the CEB code. The MOE value according to the age is obtained using the coefficient *β_cc_* in Equation (2).
(6)$$\begin{array}{c} E_{ci}\left(t\right)=\beta_{E}\left(t\right)E_{ci}\\ \beta_{E}\left(t\right)={\left[\beta_{cc}\left(t\right)\right]}^{0.5} \end{array}$$

In Equation (6), $E_{ci}$ indicates the initial MOE at 28 days.

Current design codes have a limitation in MOE prediction considering the characteristic of UHPC [17]. The considerable experimental literature had confirmed that MOEs of UHPC range from 40–50 Gpa, and equations had also been suggested [18,19,20]. However, studies on the time-dependent MOE of UHPC are rarely conducted. The time-dependent effects need to be investigated when considering the effect of MOE on structural performance.

## 3. Experiment Setup

The 150 Mpa UHPC with the characteristics summarized in Table 1 was used in this study. Fine aggregate with a density of 2.62 g/cm^3^ and an average particle size of 0.5 mm was used, and a filler with a size of 30 μm or less was mixed in UHPC. Steel fibers with a length of 13 mm and a diameter of 0.2 mm were added at a content of 1% to improve the tensile performance of UHPC.

The performance of UHPC over time was analyzed under different curing conditions. Air curing, moist curing, and steam curing were performed under the conditions listed in Table 2. For all specimens, each curing condition was maintained after steam curing for the first three days; this variable setting was performed to compare the long-term performances of UHPC stored under different conditions after 3-day steam curing.

For the compressive strength and MOE, five Φ100 mm specimens were prepared, and the average values of three specimens, excluding the maximum and minimum values, at each age were used. Three specimens were cast for flexural strength, and the long-term performance was evaluated.

### 3.1. Compressive Strength and Modulus of Elasticity

The compressive strength and MOE experiments were performed when the age of the concrete specimens reached the intended age. The compressive strength was measured using a UTM of 3000 kN capacity with a displacement control method at a 0.1 mm/s loading rate, as shown in Figure 1. Three strain gauges were installed for elastic modulus measurements. The experiments were performed on the five specimens by age based on each curing condition. The average value for the remaining three specimens was defined as the compressive strength and elastic modulus at that age, except for the two results with the most significant deviations.

### 3.2. Flexural Tensile Strength

As shown in Figure 2, a three-point flexural strength was tested using a 100 × 100 × 400 mm specimen to measure the flexural tensile strength. The experiments were conducted by adjusting the rate to 1/1500 of the span (300 mm) per minute. A linear variable displacement transducer (LVDT) with a capacity of 10 mm was used to measure the deflection of the specimen.

## 4. Compressive Strength Results

### 4.1. Experiment Results

Figure 3 shows the compressive strength results based on the age and curing method. The compressive strength at the end of steam curing was 91.2 Mpa, and the compressive strengths of each curing method were compared after three days of age. The compressive strengths for air, moist, and steam curing were 136.4, 118.0, and 170.1 Mpa at 28 days and 169.0, 190.9, and 197.1 Mpa at 365 days. The compressive strength tendency shown in Figure 3 could be attributed to the hydration of concrete. Steam curing showed high strength from the early stages because sufficient heat and moisture were supplied, so the hydration reaction was accelerated at the early stages. Steam curing typically accelerates the hydration with high temperatures and atmospheric pressure and supplies sufficient moisture for the hydration of cement. Through this process, the structure between the cement particles inside the concrete matrix becomes stronger, and the concrete under steam curing shows high strength [21,22]. The steam-cured specimens exhibited the highest compressive strength values, whereas the air-cured specimens showed the lowest values. Compared to the compressive strength of the steam-cured specimens at 365 days, the air-cured and moist-cured specimens showed strengths at 85.9 and 96.9% levels, respectively. For moist curing, the strength appeared to have consistently increased over the long term because sufficient moisture was supplied. In the case of air curing, the lowest compressive strength results were observed because the moisture and heat required for the hydration reaction were not sufficiently provided [23,24]. Compared to the strengths at 28 days, those at 365 days increased by 23.9, 61.8, and 15.9% for the air-cured, moist-cured, and steam-cured specimens, respectively, which confirmed that moist curing had the most remarkable improvement in strength over time.

### 4.2. Comparison Experiments and Prediction Model

The compressive strength results of air curing and the ACI and BS (& CEB) standards are shown in Figure 4. Comparing the tendency of compressive strength results with the prediction formula of the standards shows a high degree of agreement. When considering the application range of the standards, the compressive strength under the air curing condition was 136.4 MPa at 28 days, which significantly exceeded the range. However, since the current standard predicted the long-term compressive strength during air curing very well, using the current standard would be evaluated as appropriate.

Figure 5 shows the comparative results of moist curing. The ACI and BS & CEB standards do not consider the influence of humidity on compressive strength. Therefore, a comparison with the prediction models was performed under the air curing condition in Figure 5. The experimental values were found to be significantly higher than the results of the prediction models because the prediction models could not reflect the significant increase in strength over the long term caused by supplying sufficient humidity to the concrete. Thus, a suitable prediction model would be required to predict the long-term compressive strength of moist-cured UHPC.

Figure 6 shows the comparative results of steam curing. The ACI and BS standards provide prediction models for the compressive strength of steam-cured concrete, as shown in Equations (1) and (3); however, the CEB standard does not. Therefore, a comparison was performed using the prediction models under the general conditions of Equation (2) in Figure 6. The prediction models of ACI and CEB could be evaluated to be similar when overall tendencies were compared, and the BS showed an overestimating tendency. Since the prediction model of ACI appeared to be close to the experimental values at early ages, it tended to underestimate the compressive strength over the long term. The prediction model of CEB was similar to experimental values over the long-term, but differences at the early ages were observed. Therefore, an appropriate prediction model is required to reflect the characteristics of UHPC subjected to steam curing.

### 4.3. Proposed Prediction Model

In this study, the improving prediction model was proposed to overcome the previous limitations in the previous criteria of ACI, BS, and CEB. The tendency of each improved model was compared with that of the experimental data to select the most appropriate model. The ACI criteria in Equation (1) present coefficients a and b according to the cement type and curing method. Coefficients a and b suitable for UHPC were proposed through the analysis of the long-term behavior experiment results. The BS and CEB of Equation (2) use coefficient s, which is related to the cement type, and this coefficient have a limit of 60 MPa in the standards. Therefore, this study suggested a coefficient suitable for 150 MPa UHPC. In addition, the improvement of Equation (3) of the BS code for steam curing was investigated.

Table 3 and Figure 7 show the air curing experiment results and modified models. The modified equations of ACI and CEB(BS) were given as Equations (7) and (8).
(7)$$f_{c}\left(t\right)=\frac{t}{3.7+0.82t}f_{c}\left(28\right)$$
(8)$$f_{c}\left(t\right)=f_{c}\left(28\right)\times \exp[0.25[1-{\left(\frac{28}{t}\right)}^{\left(1/2\right)}]]$$

In the ACI model, coefficients a and b were 4.0 and 0.85, respectively, which were not significantly different compared with the modified model equation. The error was 2.9% over the long term from 90 to 365 days when the strength at each age was predicted using Equation (7). CEB and BS use the same equation under air curing conditions. Therefore, those standards could be improved to Equation (8). From the experiment results, the coefficient s was 0.25, and the error was 2.1% over the long term. In the air curing condition, the previous prediction models showed no significant difference, as shown in Figure 4, and the improved CEB & BS model was more suitable than the ACI model.

Table 4 and Figure 8 show the results under the moist curing condition. The modified equations of ACI and CEB(& BS) are respectively given in Equations (9) and (10).
(9)$$f_{c}\left(t\right)=\frac{t}{6.0+0.6t}f_{c}\left(28\right)$$
(10)$$f_{c}\left(t\right)=f_{c}\left(28\right)\times \exp[0.7[1-{\left(\frac{28}{t}\right)}^{\left(1/2\right)}]]$$

The coefficients a and b in the modified ACI model were 6.0 and 0.6, respectively, and the coefficient s in the BS & CEB model was 0.7. The errors over the long term from 90–365 days were found to be 1.2 and 2.9%, which implies that the ACI model has a minor error. From 90 days to 365 days of age, there was no significant difference between modified BS & CEB and ACI. However, since the ACI showed good agreement from 90 to 365 days than BS & CEB, the modified ACI model could be a little more suitable in moist curing conditions.

Table 5 and Figure 9 show the results for steam curing. The modified equations of ACI, CEB, and BS are given as Equations (11)–(13).
(11)$$f_{c}\left(t\right)=\frac{t}{0.8+0.47t}f_{c}\left(3\right)$$
(12)$$f_{c}\left(t\right)=f_{c}\left(3\right)\times \exp[0.88[1-{\left(\frac{3}{t}\right)}^{\left(1/2\right)}]]$$
(13)$$f_{c}\left(t\right)=f_{c}\left(3\right)+0.79\times \frac{f_{c}\left(28\right)-f_{c}\left(3\right)}{\log\left(28-t_{s}+1\right)}\log\left(t-t_{s}+1\right)$$

In these equations, the compressive strength at the end of steam curing, *f_c_*(3), was used, unlike in the air and moist curing methods. The coefficients a and b in Equation (11) were 0.8 and 0.47, respectively. The coefficient of Equation (12) was 0.88. In the BS model, *t_s_* was set to 3, which is the end time of steam curing, and the coefficient of 0.79 was used. From the comparative results, BS showed the lowest error of 2.0% for the long-term strength (90–365 days). While the ACI model described the long-term behavior of UHPC, as shown in Figure 9, the significant errors had occurred at early ages, as indicated in Table 5. The prediction model of CEB showed the lowest performance, and the most significant error could be observed for the long-term strength. From these results, the modified BS prediction model could be evaluated as the most suitable.

## 5. Modulus of Elasticity Results

### 5.1. Experiment Results

Figure 10 shows the MOE results according to age. The MOEs were 40.5, 36.2, 44.1 GPa at 28 days, and 43.8, 45.4, 45.9 GPa at 365 days. The MOE differed based on the curing method until 28 days. However, it showed no significant difference at 365 days. The MOE shows no significant difference with an increase in age because it is dominantly affected by the materials than the influence of compressive strength [17,25]. In other words, the microstructure formed by the interfacial transition zone between the binder and other materials affects the MOE more than the curing methods.

### 5.2. Comparison Experiments and Prediction Model

Figure 11, Figure 12 and Figure 13 show the MOEs according to the concrete age for air, moist, and steam curing conditions. The prediction model of the ACI, BS, and CEB standards demonstrated outstanding accuracy. Since MOE is significantly affected by the materials used, the MOE results could also be accepted as typical, considering the MOE of UHPC ranging from 40–50 GPa in the literature [26,27].

There was no significant difference in MOE depending on the curing method, as also shown in Figure 10. From 90–270 days of age, MOE was similar for the air, moist, and steam-curing specimens. Consequently, as shown in Figure 11, Figure 12 and Figure 13, prediction models of standards were found to properly predict the MOE of UHPC even though the equations did not include the influence of variables such as temperature, relative humidity, and curing method. Therefore, the MOE of UHPC at each age, which refers to the existing standards, would be considered appropriate. However, in this study, significant results were difficult to derive in determining which criterion is more suitable.

### 5.3. Proposed Prediction Model

Figure 10, Figure 11, Figure 12 and Figure 13 showed no significant difference in MOE depending on age. In general, MOE tends to increase when there is a significant increment in strength, such as normal strength concrete (NSC), high-strength concrete(HSC), and ultra-high-strength concrete(UHSC) [27,28]. In this study, the 150 MPa UHPC was investigated to determine if it shows a different tendency according to the concrete age. Figure 14 and Figure 15 show the correlation between the compressive strength and MOE. In those figures, the effect of the age was excluded, and the compressive strength and MOE were compared.
(14)$$E_{c}=a{\left(f_{c}\right)}^{b}\;\left(\mathrm{MPa}\right).$$

The previous concrete prediction models are listed in Figure 14. All the proposed equations have the form of Equation (14), and the coefficients are listed in Table 6. Most of the design criteria are designed for NSC, showing low accuracy for compressive strength of 150 MPa. Therefore, previous standards based on NSC had difficulty predicting the MOE of UHPC.

Figure 15 compares the MOE equations for UHPC by referring to the literature listed in Table 6. A coefficient proposed to define the correlation between compressive strength and MOE obtained from the results of this study was also shown. Since literature equations for predicting the MOE of UHPC are based on 28 days of age, differences occurred in specific areas. The equation proposed in this study would be expected to be applicable regardless of the age of the concrete. However, it will be necessary for sufficient experimental data on more diverse mineral admixtures to propose an equation for the MOE of UHPC.

## 6. Flexural Tensile Strength Results

### 6.1. Experiment Results

The flexural tensile strength experimental results were summarized in Table 7 and Figure 16. The results were higher for the moist and steam curing specimens than for the air curing specimens. Differences in flexural strength between 28 and 365 days of age were found to be 1.91, 19.4, and 9.84% for the air, moist, and steam curing methods, respectively. Considering that compressive strength increased by 20.8, 34.9, and 11.9%, respectively, at the same ages by referring to the compressive strength test results in Table 3, Table 4 and Table 5, flexural strength increments as high as compressive strength increments could not be expected. Since the tensile and flexural tensile strength of fiber reinforced concrete is more influenced by the distribution and directionality of fibers than the degree of hydration of concrete, these results could be confirmed that was caused by the fiber effect [32,33] Therefore, it could be determined that UHPC did not cause deterioration in flexural performance regardless of the age of the concrete, and UHPC structures such as long-span bridges would be guaranteed the life of structural systems. For air curing conditions, since the specimen showed relatively low performance than other curing methods, the careful structural design would be needed.

### 6.2. Relationship between Flexural and Compressive Strength

Figure 17 shows the correlation between the compressive and flexural strengths of concrete, ignoring the effects of age on UHPC. The flexural strength of concrete would be typically affected by fibers [34,35]. However, the experiment results showed a tendency in that the flexural strength of UHPC increased with an increase in its compressive strength. The tendency could be expressed in Figure 17. When considering that the same fiber content and mix conditions were applied in this study, an increase in compressive strength was found to contribute to the flexural strength to some extent for UHPC. Regarding these results, through studies on additional variables, including fiber contents amount and mixed material, it is expected that the correlation between the compressive and tensile strength of UHPC could be derived.

## 7. Conclusions

This study was conducted to investigate the long-term characteristics of precast UHPC members stored under different curing conditions. The obtained results would be expected to contribute to efficient member production and process management during the construction of large structures, such as super-long-span bridges. Therefore, the time-dependent effect of UHPC was investigated while focusing on curing methods.

The main variables from 3–365 days were the moist, water, and steam curing. The compressive strength, MOE, and flexural tensile strength were analyzed. Based on the experiment results, mechanical properties according to age were identified under air, moist, and steam curing conditions. The results could be summarized as follows:

There was a difference in compressive strength development characteristics based on the curing method. Steam curing exhibited the highest level of strength development, while air curing showed the lowest strength development. In the comparison of MOE, significant differences did not occur depending on the age and curing method.

The experiment results were compared with the compressive strength prediction models of ACI, BS, and CEB. The equations of the criteria properly predicted the strength at each age in the case of air curing. The prediction models had difficulty describing the strength development characteristics under the moist curing condition. The prediction models of ACI and BS for steam curing were observed differences in predicting strength development characteristics over the long term. Based on the experiment results, therefore, the improvement of the prediction model under each curing condition was suggested. As a method of modifying the coefficients of each design criterion, a prediction model for the compressive strength of UHPC was suggested. Accumulating more UHPC long-term strength data from further studies could enhance this model.

Flexural strength showed no significant difference depending on the age because steel fibers were mixed with UHPC, which significantly contributed to the flexural performance.

The correlations of MOE and flexural strength with compressive strength were derived. To predict the mechanical properties of UHPC, the model that could estimate MOE and flexural strength according to the difference in compressive strength was suggested. Based on the model, the additional research on fiber type, mixed materials, and mix proportion for UHPC could improve the completeness and expand the applicability broadly.

## Figures and Tables

**Figure 1 materials-15-05066-f001:**
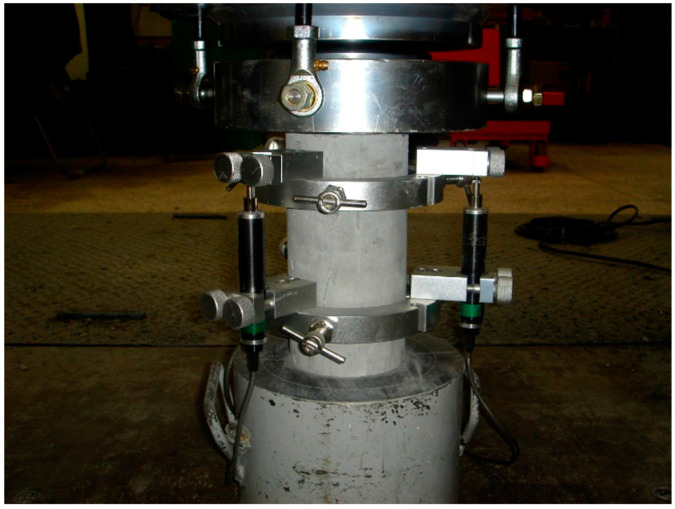
Experimental setup for the compressive strength and MOE measurements.

**Figure 2 materials-15-05066-f002:**
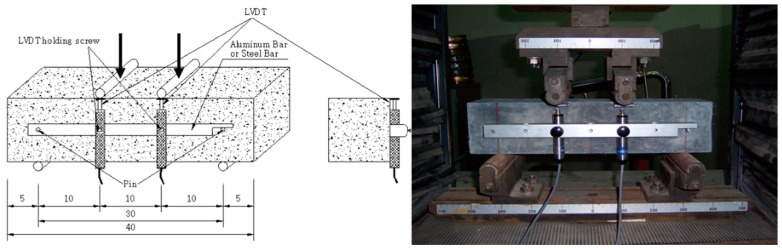
Experimental setup for the flexural tensile strength measurement.

**Figure 3 materials-15-05066-f003:**
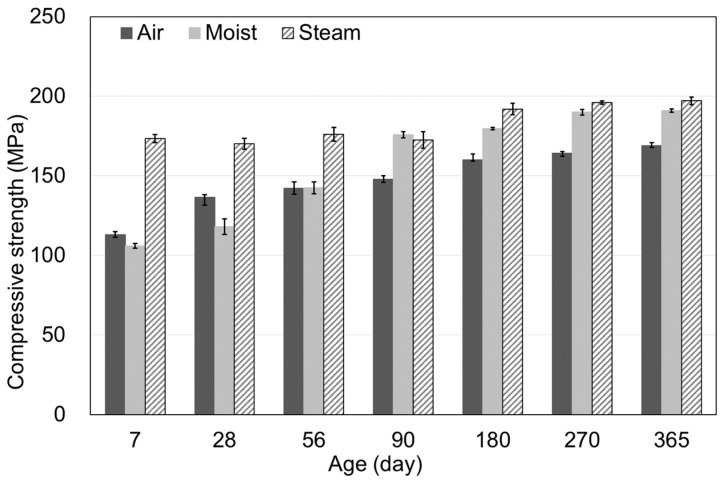
Compressive strength according to curing methods.

**Figure 4 materials-15-05066-f004:**
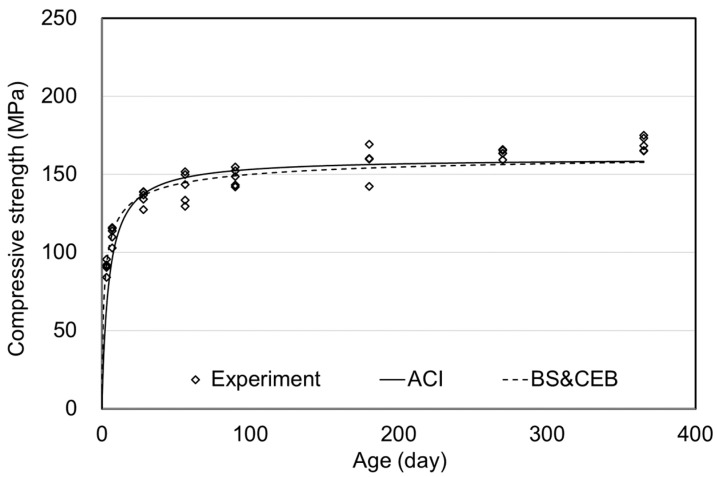
Comparative results of air curing compressive strength.

**Figure 5 materials-15-05066-f005:**
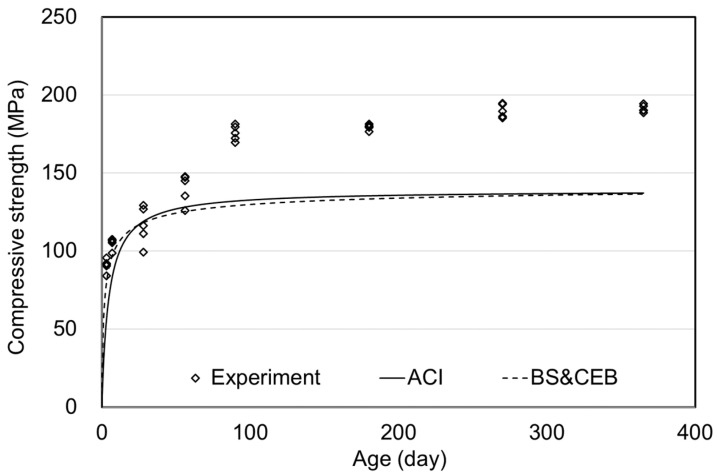
Comparative results of moist curing compressive strength.

**Figure 6 materials-15-05066-f006:**
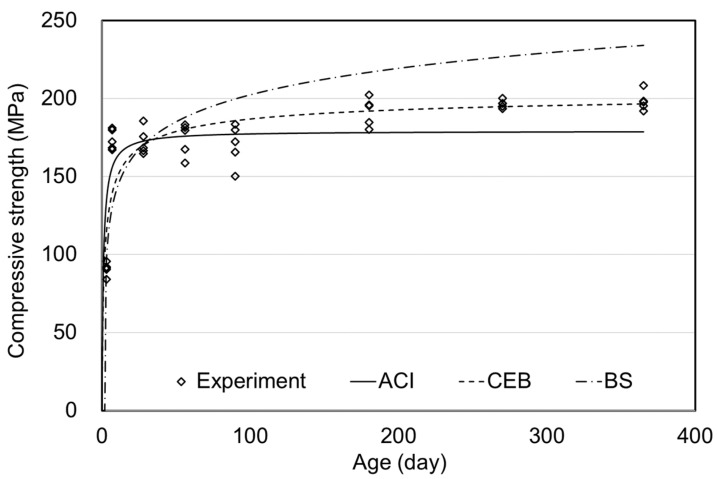
Comparative results of steam curing compressive strength.

**Figure 7 materials-15-05066-f007:**
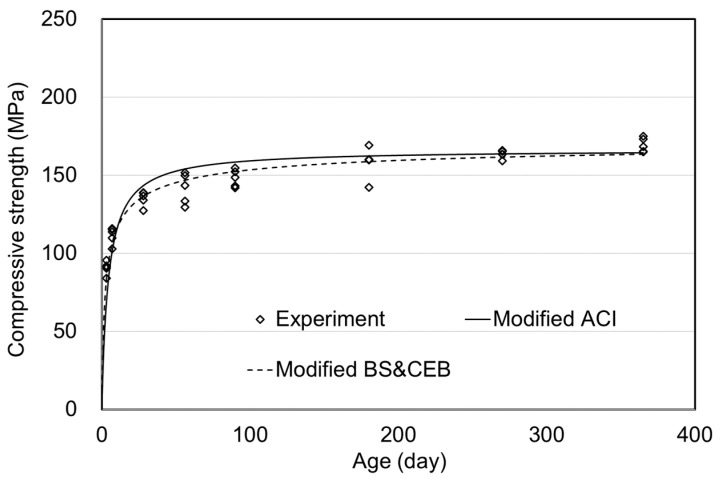
Prediction model of compressive strength for air curing.

**Figure 8 materials-15-05066-f008:**
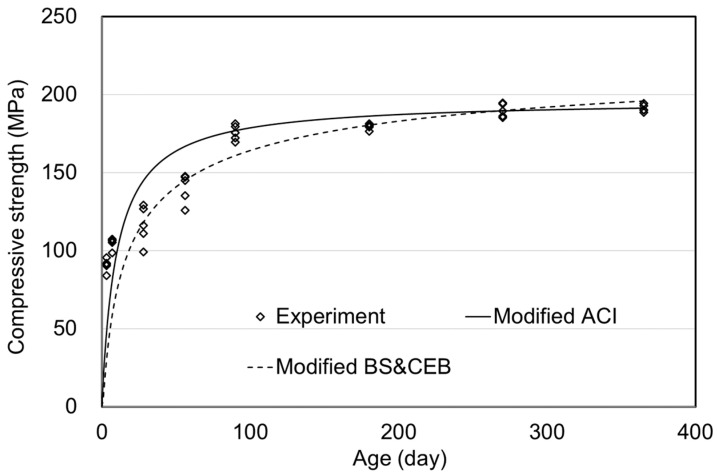
Prediction model of compressive strength for moist curing.

**Figure 9 materials-15-05066-f009:**
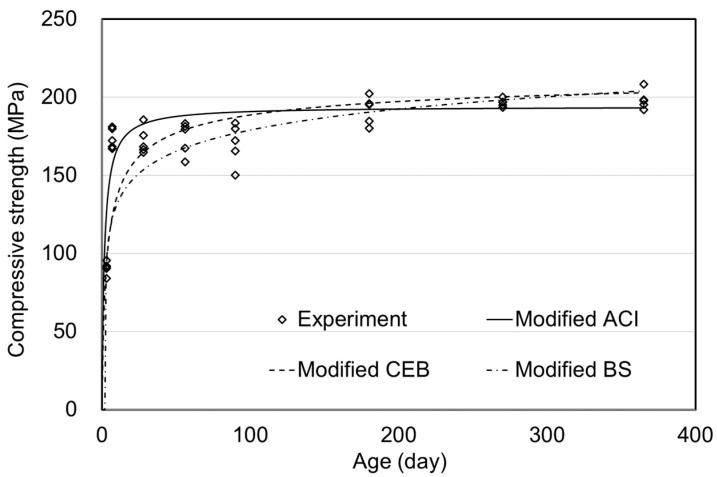
Prediction model of compressive strength for steam curing.

**Figure 10 materials-15-05066-f010:**
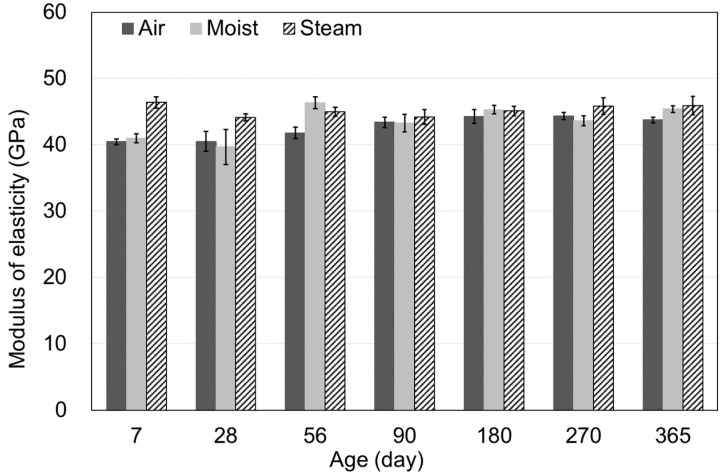
Elastic modulus according to the curing methods.

**Figure 11 materials-15-05066-f011:**
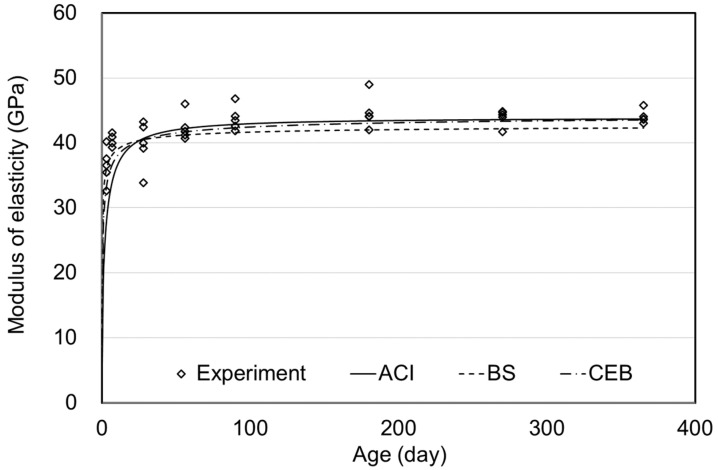
Elastic modulus of air curing.

**Figure 12 materials-15-05066-f012:**
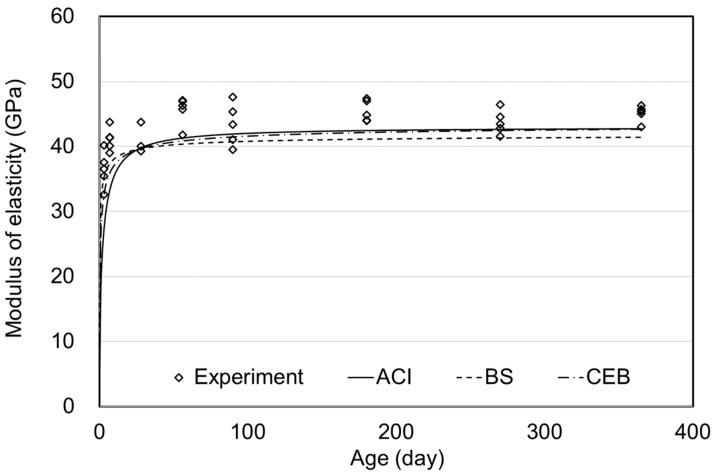
Elastic modulus of moist curing.

**Figure 13 materials-15-05066-f013:**
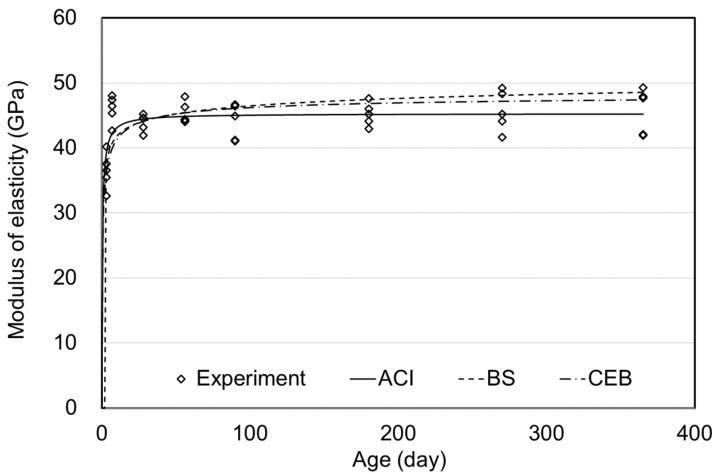
Elastic modulus of steam curing.

**Figure 14 materials-15-05066-f014:**
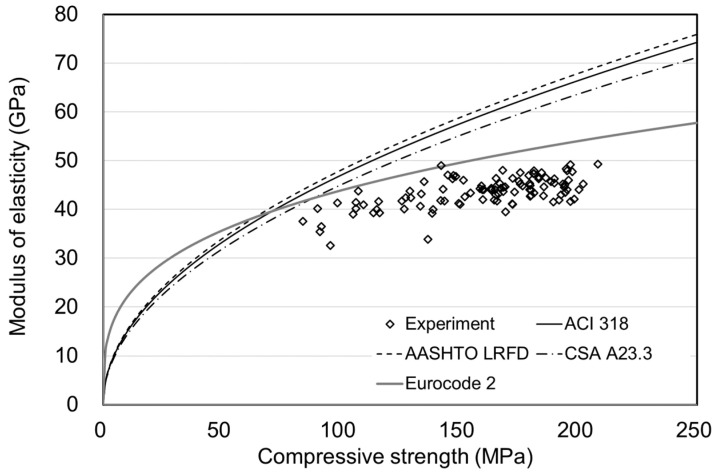
Compressive strength and MOE with the proposed equation for NSC.

**Figure 15 materials-15-05066-f015:**
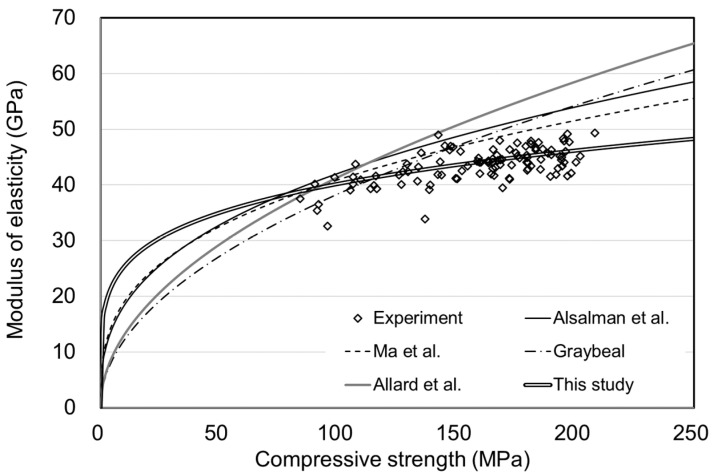
Compressive strength and MOE with the proposed equation for UHPC.

**Figure 16 materials-15-05066-f016:**
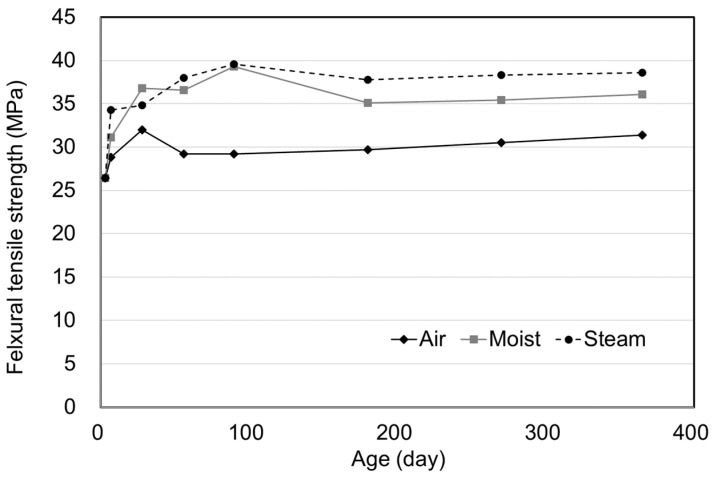
Experiment results for flexural tensile strength.

**Figure 17 materials-15-05066-f017:**
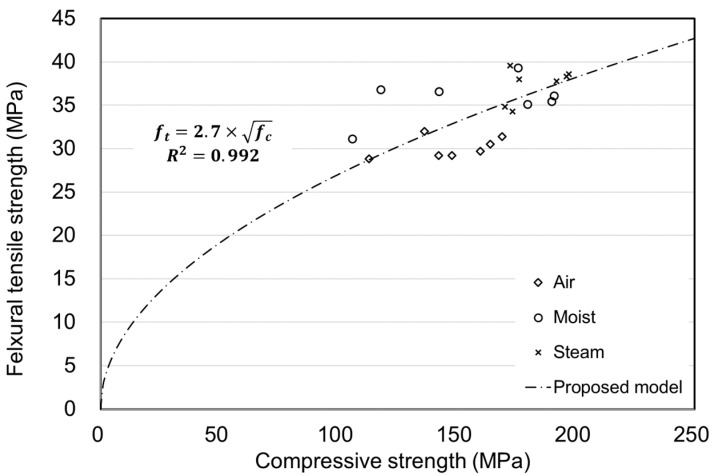
Relationship between flexural tensile and compressive strength.

**Table 1 materials-15-05066-t001:** Mechanical and chemical characteristics for UHPC.

Materials	Surface Area (cm^2^/g)	Density (g/cm^3^)	Ig.Loss (%)	Chemical Composition (%)					
				SiO_2_	Al_2_O_3_	Fe_2_O_3_	CaO	MgO	SO_3_
OPC	3413	3.15	1.40	21.01	6.40	3.12	61.33	3.02	2.3
S.Fume	200,000	2.10	1.50	96.00	0.25	0.12	0.38	0.10	–
Filler	–	–	0.01	99.3	0.15	0.01	0.03	0.004	–

**Table 2 materials-15-05066-t002:** Variables for experiments and curing conditions.

Specimens	Size (mm)	Number of Specimens	Curing Methods	Age
Compressive strength and modulus of elasticity	Φ100 × 200	5	Air curing (23 °C, 50%, 28 days)	3, 7, 28, 56, 90, 180, 270, 365
Flexural tensile strength	100 × 100 × 400	3	Moist curing (23 °C, 100%, 28 days)	
			Steam curing (90 °C, 100%, 3 days)	

**Table 3 materials-15-05066-t003:** Experiment results in air curing compared with the modified prediction models.

Age	Average Results of Experiment (MPa) (STD)	Modified ACI (MPa)	Modified CEB & BS (MPa)
7	112.9 (2.4)	101.1	106.2
28	136.4 (2.6)	143.2	136.4
56	142.3 (5.6)	153.9	146.7
90	148.0 (3.3)	158.4	152.3
180	159.9 (6.2)	162.3	158.7
270	164.1 (1.7)	163.6	161.6
365	169.0 (3.1)	164.3	163.4
Error (90–365 days)		2.9%	2.1%

**Table 4 materials-15-05066-t004:** Experiment results in moist curing compared with modified prediction models.

Age	Average Results of Experiment (MPa) (STD)	Modified ACI (MPa)	Modified BS and CEB (MPa)
7	106.0 (1.5)	81.0	58.6
28	118.0 (5.7)	144.9	118.0
56	142.4 (5.4)	166.9	144.9
90	175.8 (3.4)	177.0	160.8
180	179.7 (1.3)	186.3	180.3
270	190.0 (3.3)	189.7	189.7
365	190.9 (1.8)	191.5	195.8
Error (90–365 days)		1.2%	2.9%

**Table 5 materials-15-05066-t005:** Experiment results in steam curing compared with modified prediction models.

Age	Average Results of Experiment (MPa) (STD)	Modified ACI (MPa)	Modified CEB (MPa)	Modified BS (MPa)
7	173.4 (4.5)	156.0	123.5	122.0
28	170.1 (5.8)	182.8	164.8	153.5
56	176.1 (7.5)	188.2	179.3	167.5
90	172.5 (9.1)	190.4	187.2	176.9
180	191.9 (6.9)	192.1	196.2	190.4
270	196.1 (2.0)	192.7	200.3	198.2
365	197.1 (4.8)	193.1	202.9	204.0
Error (90–365 days)		3.6%	4.0%	2.0%

**Table 6 materials-15-05066-t006:** Coefficient of Equation (14) for the proposed equations for MOEs.

Committee or Literature	a	b	Range of Compressive Strength (MPa)
ACI 318	4700	0.5	<41
AASHTO LRFD *	4800	0.5	<105
CSA A23.3 *	4500	0.5	<40
Eurocode 2	11,025	0.3	<98
Ma et al. [18]	8820	1/3	150–180
Alsalman et al. [29]	8010	0.36	31–235
Graybeal [30]	3840	0.5	124–193
Allard et al. [31]	4139	0.5	22–210
This study	16,000	0.2	84–208

* AASHTO: American Association of State Highway and Transportation Officials. LRFD: Load Resistance Factor Design CSA: Canadian Standards Association.

**Table 7 materials-15-05066-t007:** Experiment results for flexural tensile strength.

Age	Air Curing (MPa) (STD)	Moist Curing (MPa) (STD)	Steam Curing (MPa) (STD)
3	26.4 (1.5)	26.4 (1.5)	26.4 (1.5)
7	28.8 (1.6)	31.1 (0.1)	34.3 (2.0)
28	32.0 (1.2)	36.8 (1.4)	34.8 (0.6)
56	29.2 (1.5)	36.6 (1.0)	38.0 (2.2)
90	29.2 (2.0)	39.3 (2.1)	39.6 (1.4)
180	29.7 (1.9)	35.1 (0.2)	37.8 (1.0)
270	30.5 (0.7)	35.4 (0.8)	38.3 (0.9)
365	31.4 (0.7)	36.1 (0.8)	38.6 (0.6)

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
