# Peer review of "The Time-Dependent Effect in Ultra High-Performance Concrete According to the Curing Methods"

_materials, 2022, doi:10.3390/ma15145066_

Round 1
Reviewer 1 Report
This paper addresses a practical problem for efficient and sustainable precast construction –concrete members exhibit significant differences over time as they would be stored off-site for days under different conditions after the completion of steam curing, especially for ultra-high-performance concrete (UHPC) members. The authors proposed a series of experiment testing that compares the performance of UHPC under air, moist, and steam storage conditions after 3-day steam curing, respectively. Based on the experimental results, the development characteristic of compressive strength, modulus of elasticity (MOE), and flexural strength until 365 days of age were generalized. The experimental results presented that only the development characteristic of compressive strength which is closely related to on the degree of hydration showed a significant difference under different curing conditions, and suggested that MOE and flexural strength could be estimated according to the compressive strength but regardless the difference in curing methods and age. The models, which were proved to be more reliable in this study, were then derived for predicting the above mechanical properties of UHPC after comparing the results with the prediction models of specifications or literatures. Those proposed formulas could provide references for the additional research on some significant variables, such as fiber type, mineral admixtures, and mix pro-portion, to expand the applicability.
Overall, the article is well organized and its presentation is good. However, some minor issues still need to be improved:
(1) Please check the typos such as “ultrahigh-performance concrete” in Line 29.
(2) Line 48 to 55, Line 214 to 218, Line 250 to 251, Line 283 to 285 & Line 365 to 367: please check the grammar mistakes and improve the English writing.
In general, there is a lack of explanation of replicates and statistical methods used in the study. Please improve the explanation. An example is shown in line 206 to 208 & line 331: the points illustrated in the original texts “The MOE shows no significant difference with an increase in age because it is dominantly affected by the materials than the influence of compressive strength.” and “However, the MOE of concrete increases with an increase in the compressive strength.” are inconsistent.
Reviewer 2 Report
1- in abstract, put some percentage difference of strength with curing. Show some results in number form.
2-in the intro section, add some literature regarding effect of curing on uhpc. Also add some review papers on uhpc.
3- add novelty and originality of this study conducted.
4-can u put pictures of failed specimens and crack pattern.
5- add a separate section for discussion of results. Also compare your findings with other researchers .
6- references need to follow journal guidelines
Reviewer 3 Report
Authors investigated the time-dependent effect in UHPC according to the curing methods. The presentation of the paper is good. I recommend this paper for minor revision.
(1) Avoid abbreviations in title of the manuscript.
(2) Include some numerical values in the abstract, example the strength values.
(3) Please cite some references in the introduction section.
(4) If possible merge introduction and literature survey sections.
(5) Table 1 – chemical composition at what percentages?
(6) …….average particle size of 0.5 mm- how particle size was measured? Provide details.
(7) Why no error values are not included for the property values in Tables?
(8) How the Variables for experiments and curing conditions were selected? Please justify.
(9) Check the language of the paper.
(10) Why higher compressive strength for steam was observed? Need more explanation with suitable references.
Reviewer 4 Report
The paper: The time-dependent effect in UHPC according to the curing methods
Authors Kwangmo Lim, Kyongchul Kim, Kyungtaek Koh, and Gumsung Ryu presents an interesting topic to Materials readers, more corrections are necessary.
My principal questions or remarks:
The title is clear.
The content is in accord with title.
The manuscript not adhere to the journal's standards in this form.
The size of the article is appropriate to the contents.
The authors must underline the major findings of their work and explain novelty of this study. The objective must be better pointed.
The Abstract section refers to the study findings, methodologies, discussion as well as conclusion.
The key words permit found article in the current registers or indexes.
In the introduction isn’t clearly described the state of the art of the investigated problem. In Introduction the author hasn’t any references. The authors can present 2. Literature review in 1. Introduction.
All equations must be revised and explain the notations.
The methods are well described.
The authors can restructure manuscript: 3. Results and discussion… and 3.1. Compressive strength and modulus of elasticity; 3.2. Flexural tensile strength….
The figures have a good quality. Figure 6 is not very clear… please explain.
In Figure 15. Compressive strength and MOE with the proposed equation for UHPC… the legend can be write after figure because [00]… [00] is not good… or write the number of reference.
The tables contain necessary results.
The Conclusion is OK. The main results are presented in this section.
Please present full form of some abbreviations when they first appeared. Please respect for cement the abbreviation presented at row 49, or change the notations.
The references aren’t in journal’s format.
The paper isn’t easy to understand by readers from other area. For example, AASHTO LRFD is hard to understand by other researchers, etc.
6. Patent…Is necessary?
The paper was written in standard, grammatically correct English, small corrections are necessary.
The manuscript must be restructured and carefully verified. The results are very interesting and must be valorized and presented very clear.
Round 2
Reviewer 2 Report
Addresses most of the previous comments. References formatting still need revision which may be adjusted later-on.
Reviewer 4 Report
The manuscript was improved in accord with recommendations.